# Experimental Investigation of the Biofunctional Properties of Nickel–Titanium Alloys Depending on the Type of Production

**DOI:** 10.3390/molecules27061960

**Published:** 2022-03-17

**Authors:** Minja Miličić Lazić, Peter Majerič, Vojkan Lazić, Jelena Milašin, Milica Jakšić, Dijana Trišić, Katarina Radović

**Affiliations:** 1School of Dental Medicine, University of Belgrade, 11000 Belgrade, Serbia; minja.milicic@stomf.bg.ac.rs (M.M.L.); vojkan.lazic@stomf.bg.ac.rs (V.L.); jelena.milasin@stomf.bg.ac.rs (J.M.); milica.jaksic@stomf.bg.ac.rs (M.J.); dijana.trisic@stomf.bg.ac.rs (D.T.); 2Faculty of Mechanical Engineering, University of Maribor, 2000 Maribor, Slovenia; peter.majeric@um.si

**Keywords:** nickel–titanium, continuous casting, characterization, biofunctional properties, biocompatibility

## Abstract

Nickel–titanium alloys used in dentistry have a variety of mechanical, chemical, and biofunctional properties that are dependent on the manufacturing process. The aim of this study was to compare the mechanical and biofunctional performances of a nickel–titanium alloy produced by the continuous casting method (NiTi-2) with commercial nitinol (NiTi-1) manufactured by the classical process, i.e., from remelting in a vacuum furnace with electro-resistive heating and final casting into ingots. The chemical composition of the tested samples was analyzed using an energy dispersive X-ray analysis (EDX) and X-ray fluorescence (XRF). Electron backscatter diffraction (EBSD) quantitative microstructural analysis was performed to determine phase distribution in the samples. As part of the mechanical properties, the hardness on the surface of samples was measured with the static Vickers method. The release of metal ions (Ni, Ti) in artificial saliva (pH 6.5) and lactic acid (pH 2.3) was measured using a static immersion test. Finally, the resulting corrosion layer was revealed by means of a scanning electron microscope (SEM), which allows the detection and direct measurement of the formatted oxide layer thickness. To assess the biocompatibility of the tested nickel–titanium alloy samples, an MTT test of fibroblast cellular proliferation on direct contact with the samples was performed. The obtained data were analyzed with the IBM SPSS Statistics v22 software. EDX and XRF analyses showed a higher presence of Ni in the NiTi-2 sample. The EBSD analysis detected an additional NiTi_2_-cubic phase in the NiTi-2 microstructure. Additionally, in the NiTi-2 higher hardness was measured. An immersion test performed in artificial saliva after 7 days did not induce significant ion release in either group of samples (NiTi-1 and NiTi-2). The acidic environment significantly increased the release of toxic ions in both types of samples. However, Ni ion release was two times lower, and Ti ion release was three times lower from NiTi-2 than from NiTi-1. Comparison of the cells’ mitochondrial activity between the NiTi-1 and NiTi-2 groups did not show a statistically significant difference. In conclusion, we obtained an alloy of small diameter with an appropriate microstructure and better response compared to classic NiTi material. Thus, it appears from the present study that the continuous cast technology offers new possibilities for the production of NiTi material for usage in dentistry.

## 1. Introduction

Shape memory alloys (SMAs), thanks to their unique thermomechanical function, belong to the group of functional materials that is manifested through the effects of shape memory and the superelasticity. Based on the primary alloying elements, shape memory alloys can be classified into three groups: Cu-based, Fe-based, and nickel–titanium (commercial name nitinol) alloys. From these, nitinol is particularly popular in dental devices thanks to its larger recoverable strain, great corrosion resistance, and better biocompatibility [1,2]. It has widespread use in dentistry for endodontic instruments for root canal procedures, orthodontics archwire, attachments in prosthodontics, and implants and plates for immobilizing fragments in maxillofacial orthopedic surgery [3,4,5,6].

The biofunctionality of nitinol is proven through its great functional properties in the biological environment. Some of these are good thermal deployment, fatigue resistance, exceptionally high wear, a non-ferromagnetic property, MR compatibility, low Young′s module, and compressive strength close to that of natural bone [7]. Strong mechanical properties ensure that this alloy can exert cyclic forces.

However, binary nickel–titanium alloys are variable in their composition, so functional characteristics are composition dependent [7]. Their higher nickel content results in a superelasticity effect; a phenomenon that makes these alloys attractive to the medical field [7,8]. Additionally, by increasing the nickel content the transformation temperature (TT) decreases and can be adapted to the temperature of the human body.

Nickel-rich alloys are still controversial in terms of their biocompatibility if long-term implantation in a biological environment is required. Regarding the different applications in the human body, the biocompatibility of nickel–titanium is contextual. Hence, its potential cytotoxicity must be observed in the target tissue and/or cells that the material will interfere with. When it comes to dentistry, the oral cavity represents a specific biological environment. Dental materials are faced with dynamic conditions, changes in acid-base status, and constant immersion in tissue fluids, as well as exposure to continuous cyclic stress due to the action of chewing forces. As a consequence of their constant immersion in tissue fluids, corrosion can occur.

It is well documented that the biocompatibility of nitinol is dependent on the release of corrosion degradation products [9,10]. Nickel–titanium alloys belong to the group of passive alloys with the ability to create a surface oxide layer that prevents further corrosion. In the literature-based evidence there is no consensus about the corrosion resistance of these alloys [9,11,12]. The characteristics of the oxide layer can vary in thickness and homogeneity. The poor quality of an oxide layer can lead to the release of toxic elements.

Considering the safety application of these alloys, a great health concern about dental devices is related to nickel toxicity, given the fact that nickel can cause a cell-mediated delayed hypersensitivity response (type IV) [9]. An extensive immune reaction is manifested clinically as contact dermatitis, pustulosis palmoplantar, lichen planus, dyshidrotic eczema, and burning mouth syndrome [9,13,14]. To ensure safe application, nickel ion release should be minimized and the key component in achieving this is the enhanced corrosion resistance of the dental device [7,12].

The main consideration concerning the fabrication of nickel–titanium alloys is related to finding an optimal manufacturing process that will ensure a nickel-free zone on the upper layers of the material. If the primary fabrication methods cannot produce high-quality surfaces, these alloys may be subjected to specific surface treatments [15,16]. Moreover, one should be aware that the most common fabrication methods for these alloys are difficult and/or expensive [17,18]. The main issues related to conventional fabrication methods are the chemical reactivity at high temperatures and the inadequate surface microstructure which affects nickel leaching [15,18,19]. The term “conventional” refers to production methods based on melting the bulk or the powder of metals and casting them into ingots. Conventional fabrication of nitinol involves vacuum arc remelting and vacuum induction melting and conventional sintering of powder metallurgy. In implant-prosthodontics, castable dental implants and abutments have been used since the 1980s.

Contrary to conventional fabrication, additive manufacturing techniques (Selective Laser Melting SLM, Selective Laser Sintering SLS, Laser Engineered Net Shaping LENS, and Electron Beam Melting EBM) are modern methods that enable the production of nickel–titanium alloys’ various shapes with highly controlled compositions [17,20]. Due to the expensive equipment and production costs these techniques have not found a wide commercial application; therefore, there is room for the improvement of the manufacturing processes, especially for usage in dentistry. After obtaining a semi-finished product, nickel–titanium alloys must be subjected to wide-ranging and extensive machining procedures to achieve the appropriate shape of devices and instruments for use. It is well known that the conventional machining processes of difficult-to-machine (DTM) materials, such as drilling, milling, and mechanical cutting, are very difficult to perform due to high ductility, the work hardening of the object, and significant tool wear. Several authors give special importance to the methods of non-contact electro erosive processing (EDM) [21,22,23,24]. This method uses thermoelectric energy for the removal of the material, providing a higher quality of finished surface. It gives the possibility of creating a “nickel-free zone” within the upper surface, which consists mainly of TiO_2_ and small amounts of NiO and Ni_2_O_3_ [9]. It has not been reported that this method can cause a negative impact on the functional properties of the material.

Considering the fact that any material used for dental purposes must combine certain mechanical, chemical, and biofunctional properties, those features are investigated within the new production model for nickel–titanium alloys.

This paper analyzes the properties of nickel–titanium alloys obtained via two different production processes, classical and continuous casting. Classical casting consists of remelting in a vacuum furnace with electro-resistive heating and a final casting into various ingots. After casting, an alloy is subjected to thermo-mechanical treatment processes (rolling or drawing) where large deformations occur. Thermo-mechanical treatment is performed to achieve the required dimensions of the NiTi material (rods, wires, etc.). Contrary to that, the continuous casting technology is a combination of vacuum induction remelting and vertical continuous casting. The melt continuously flows into the mold (applied with a relatively small diameter) where it solidifies. This is a great advantage that minimizes time and additional secondary procedures for processing the strands into wires, attachments, and other semi-finished products.

The aim of this study was to investigate the influence of alloy fabrication (classical or continuous casting) on the biofunctional properties of nickel–titanium alloy in terms of hardness, ion release from the surface in a testing medium, oxide layer formation, and biocompatibility.

## 2. Material and Methods

### 2.1. Sample Preparation

The materials used in the present study were nickel–titanium alloys, divided into two groups of samples. The first group, named NiTi-1 samples, consisted of commercially available market samples purchased from Merkur d.d. (Celje, Slovenia). Alloys were manufactured by the classical process, i.e., from remelting in a vacuum furnace with electro-resistive heating and a final casting into various ingots (as written on the supplier’s statement).

The second group of samples, NiTi-2, were produced by continuous casting in the form of rods, according to the briefly described methodology [17]. The experimental device for obtaining NiTi-2 alloys consists of a vacuum induction melting (VIM) furnace and a vertical continuous caster. The appropriate amount of solid alloy (about 20 kg) was put into a crucible and melted in a vacuum induction furnace (10-2 mbar). VIM was performed at a medium frequency (8000 Hz). The temperature of casting was around 50 °C above the temperature of melting for NiTi alloys, which is 1350 °C. The melt continuously flowed under an Ar atmosphere into the mold cooled by water where it solidified.

All samples were cut by electro-erosion (EDM): (i) the NiTi-1 was cut into square shapes (with slide length 10 mm, and thickness 1.2 mm)and (ii) the NiTi-2 into shaped discs (ø = 11 mm, thickness 1.6 mm) suitable for further metallographic preparation and testing. The metallographic preparation included mounting in a hot-mounting mass and grounding with abrasive paper in grades of 180–4000 on the grinding/polishing machine BUEHLER Automet 250, and EcoMet 250, (Lake Bluff, IL, USA). Samples were polished on the same devices with a Naples cloth and 1μm polishing suspension. After polishing, samples were cleaned with acetone, alcohol, and deionized water in ultrasound. For microstructure observations all samples were etched according to the protocol (ASTM Standard E 407, number 192 “Kroll’s”) as written below:NiTi-1—3 mL HF, 6 mL HNO_3_, 100 mL—etching time 30 s;NiTi-2—3 mL HF, 6 mL HNO_3_, 100 mL—etching time 30 s.

A metallographic characterization was obtained to determine the NiTi grain size.

### 2.2. The Determination of Chemical Composition and Phase Distribution by Scanning Electron Microscopy (SEM)

Qualitative and quantitative analyses of the samples’ composition of NiTi-1 and NiTi-2 were performed with X-ray fluorescence (XRF) analysis, using Thermo Scientific Niton XL3t GOLDD equipment.

A scanning electron microscope (SEM), Sirion 400NC (FEI, Hillsboro, OR, USA), with an energy-dispersive X-ray spectroscope (EDX), INCA 350 (Oxford Instruments, Oxford, UK), was used for the detailed microchemical analyses. The SEM/EDX measurements were as follows:(i)For NiTi-1: In the central part of the sample, two segments were chosen. In the first segment, the measurement was performed at five measuring points, and in the second segment at four points. In the marginal part of the sample, measurement was performed at four points of the measuring segment.(ii)For NiTi-2: In the central part of the sample three measuring segments with three measuring points were chosen. In the marginal part there were two measuring segments, one with four and one with two measuring points.

Detailed microstructure investigation (the visibility of the different phases of NiTi-1 and NiTi-2 samples) was performed using SEM thermal field emission (SEM JEOL JSM-6500F), equipped with electron backscatter diffraction (EBSD) analytical techniques (IMT, Ljubljana, Slovenia). Secondary-electron images and backscattered electron images were recorded at different magnifications and SEM working parameters of a 15 kV voltage, 7 nA probe current and 10 mm working distance.

### 2.3. Hardness

In order to determine the mechanical properties, the hardness of the samples was measured by the static Vickers method. Indentation hardness was measured due to the procedure corresponding to the ISO 6507-1:2018 Standard. The hardness, HV50, was measured on each specimen from each experimental group on a WPN HPO 250 machine that applied the nominal value of a 49.03 N test force load for 15 s. In this method, a diamond tip in the form of a regular four-sided pyramid was used as an indenter, in which the opposite sides overlap the angle of (136 + 1)°. An overview of the hardness imprints was examined on a Nikon EPIPHOT 300 microscope (Mellvile, NY, USA).

### 2.4. Immersion Testing and ICP-MS Analysis

A static immersion test was conducted to investigate the potential release of metal ions from both NiTi-1 and NiTi-2 samples’ surfaces. Two different mediums with different pH values were used: a solution of artificial saliva (with a composition of 1.5 gL^−1^ KCL; 1.5 gL^−1^ NaHCO_3_; 0.5 gL^−1^ NaH_2_SO_4_ × H_2_O; 0.5 gL^−1^ KSCN; and 0.9 gL^−1^ lactic acid) with a 6.5 pH value was used, corresponding to the oral environment. In order to simulate a corrosive environment, a solution containing 5.85 gL^−1^ NaCl + 10 g L^−1^ lactic acid was also used, with the pH adjusted to 2.3 with 0.1 M NaOH, according to ISO 10271 for testing materials in dentistry. Both groups of samples, NiTi-1 and NiTi-2,were immersed in 5 mL of each test solution in a test tube clogged with a rubber cork using nylon string. The test duration was 168 h at a constant temperature of 37 ± 0.2 °C. Parallel with the samples, the pure solution “zero samples” were treated in the same way. To evaluate the migration of ions from the samples into the solution, the chemical content of the suspensions was overseen by ICP-MS.

### 2.5. FIB Cross-Section of the Immersed Samples

After the immersion test was performed, the formed corrosion layers on all samples were characterized with the focus ion beam (FIB) technique. A Quanta 200 3D electron microscope with anion beam gun was used for measuring the depth of the formatted corrosion layer. A focused beam of primary gallium ions enabled the etching of the surface of the samples and the cutting of a cross-section without contamination, thus gaining a direct insight into the resulting corrosion layer. The thickness of the formatted surface oxide layer of the NiTi-1 and NiTi-2 samples was measured in 4 segments, and 5 measurement points were chosen within each of the respective segments, followed by statistical processing of the results at the end of the process.

### 2.6. In Vitro Determination of Biocompatibility

An MTT test was performed to evaluate the biofunctionality and biocompatibility testing of the NiTi-1 and NiTi-2 samples. Human gingival tissues were obtained with written consent from healthy donors, and the gingival tissue was minced into 1 mm^3^ fragments and subjected to the outgrowth method. The minced tissue was placed in 25 cm^2^ culture flasks with a growth medium (DMEM/F12 supplemented with 10% FBS and 1% ABAM, all from Gibco, Thermo Fisher, MA, USA) and incubated at 37 °C in a humidified 5% CO_2_ atmosphere. The cells were passaged regularly upon reaching 80% confluence. The culture medium was changed every 2–3 days. After the third passage, HGCs were used in the study.

For the assessment of mitochondrial activity after direct exposure to the tested materials the medium was discarded and a medium containing 3-(4,5-dimethylthiazol-2-yl)-2,5 diphenyltetrazolium bromide (MTT, 0.5 mg/mL) (Sigma-Aldrich, St. Louis, MO, USA) was added to each well and incubated. After 4 h the supernatant was discarded and dimethyl sulfoxide (Sigma-Aldrich, St. Louis, MO, USA) was added to each well. The plate was placed on a shaker for 20 min at 250 rpm, in the dark, at 37 °C. The extracted colored solutions from 12-well plates were transferred into a new 96-well plate. The optical density was measured at 550 nm using a microplate reader, RT- 2100c (Rayto, Shenzhen, China). As a control, cells were seeded onto sterile glass discs, identical in size and shape to the experimental discs. The percentage of mitochondrial activity was calculated as the difference to the control group.

Preparation of the samples for SEM observations:

After 7 days, the cells were fixed in 2% glutaraldehyde for 2 h at 40 °C, dehydrated with increasing concentrations of ethanol (30%, 50%, 70%, 90%, 100%) with 10 min for each concentration, transferred to a critical point dryer for 30 min, and gold-coated before a scanning electron microscopic evaluation in a JEOL JSM-6610LV machine (Dearborn, MI, USA).

### 2.7. Statistical Analysis

The data were analyzed using IBM SPSS Statistics v22 software (SPSS Inc., Chicago, IL, USA). An independent samples *t*-test was used for evaluating the differences in the hardness between the NiTi-1 and NiTi-2 samples.

The results of the formatted oxide layers’ thickness were shown as quantitative data and as mean ± standard deviation (SD). A quantile-quantile (Q-Q) graphical technique was used to determine if the data were normally distributed.

For the purpose of comparing the results of cell viability, an independent samples *t*-test was used for evaluating the differences between the NiTi-1 and NiTi-2 samples. A one-way analysis of variance (ANOVA) test was used for evaluating the differences between both the NiTi samples and the control. A *p*-value less than 0.05 was considered to be statistically significant.

## 3. Results

### 3.1. Chemical Composition (SEM/EDX and SEM/XRF Analyses)

The results of the EDX and XRF analyses revealed that the presence of nickel was higher in the NiTi-2 samples. Contrary to that, both analyses demonstrated that the presence of titanium was higher in the NiTi-1 sample. The analyses detected iron only in the NiTi-2 sample (Table 1).

#### 3.1.1. Phase Identification (EBSD Analysis)

The phase distribution in the microstructure of the classical cast and continuous cast samples was analyzed using EBSD as shown in Figure 1 and Figure 2. The results demonstrated the presence of NiTi-cubic and Ni_3_Ti-hexagonal phases in the NiTi-1 sample. On the other hand, an additional NiTi_2_ cubic phase was detected in the NiTi-2 sample.

#### 3.1.2. Grain Size Measurement

The microstructure was examined with optical microscopy (Figure 3) on the 500 um scale with a magnification of 50, and the grain size was determined microscopically according to the ASTM Standard. The microstructural ASTM analysis showed that, for the NiTi-1 alloy, the grain number (G) was 5 and there were 256 grains per 1 mm^2^, which indicates a typical deformed state, while in NiTi-2 the grain number was 7 and there were 1.024 grains per 1 mm^2^. The ASTM grain number increased with the decreasing grain size (Table 2).

### 3.2. Sample Hardness

The descriptive statistics for the hardness measurements are given in Table 3. The results obtained on eight measurements showed that the mean hardness value for the NiTi-2 sample was two times higher (624 HV) than for NiTi-1 (317 HV). The results of the hardness measurements obtained by the independent samples *t*-test by group (NiTi-1, NiTi-2) showed that there was a statistically significant difference in the hardness between NiTi-1 and NiTi-2 (*p* ≤ 0.001).

Figure 4 presents the optical microstructure of indentation where the length of the diagonal of the impression was created as a negative (copy) of the indenter. As it can be seen, the impression of NiTi-2 is smaller than in the NiTi-1sample.

### 3.3. ICP Analysis of Solutions after Immersion Testing

Table 4 shows the presence of released ions, measured by the ICP-MS analysis, for both NiTi-1 and NiTi-2 samples in the solutions of artificial saliva and lactic acid. Immersion in a solution of artificial saliva with a pH 6.5 after 7 days did not induce significant ion release in neither of samples (NiTi-1 and NiTi-2). Al and Cu were not detected. Moreover, it should also be noted that Fe, although present in NiTi-2, was not detected in the solutions, i.e., there was no release from the sample surface. The concentration of Ni ions was slightly higher in the solution with the NiTi-1 samples (0.05 µg/cm^2^) in comparison to the solution with the NiTi-2 samples (0.04 µg/cm^2^). The concentration of Ti ions was below the detection limit in both groups.

In the acidic environment, the suppression of Al ions was lower in the NiTi-1 alloy (0.01 µg/cm^2^) than in the NiTi-2 alloy (0.06 µg/cm^2^). Considering the migration of the Cu ions from the samples, the NiTi-1 samples released lower values (0.37 µg/cm^2^) as opposed to NiTi-2 (0.045 µg/cm^2^). The concentration of Ni ions was two times higher in the solution with the classical cast alloy NiTi-1 (2.33 µg/cm^2^) than the corresponding values in the continuous cast alloy NiTi-2 (1.2 µg/cm^2^), despite the fact that the content of nickel in the NiTi-2 alloy was higher. As far as Ti ions are concerned, the suppression was highly reduced in the NiTi-2 samples (0.63 µg/cm^2^) in comparison to NiTi-1 (2.17 µg/cm^2^).

### 3.4. FIB Cross-Section Analysis

The descriptive statistics of the thickness of the oxide layer that formed on the samples’ surfaces after the immersion testing are presented in Table 5. The results obtained on twenty measurements showed almost the same mean values (34 nm for NiTi-1 and 35 nm for NiTi-2), while the oxide layer thickness ranged between 27–55 nm for NiTi-1 and between 25–50 nm for NiTi-2.

The NiTi-2 samples showed more homogeneous data than NiTi-1 regarding the depth of the oxide layer (Figure 5).

### 3.5. Biocompatibility Results

After 24 h of direct exposure of human gingival cells (HGCs) to the tested materials, increased mitochondrial activity was observed in both tested groups (NiTi-1 and NiTi-2) in comparison to the untreated cells (the control group) (Figure 6). Higher activity was observed in cells seeded directly on the NiTi-1 samples but without statistically significant differences between NiTi-1 and NiTi-2 (*p* = 0.069). Comparing the cells’ mitochondrial activity between groups (the control group, the cells seeded on NiTi-1 and the cells seeded on NiTi-2 samples), the ANOVA test showed that there was no statistically significant increase of mitochondrial activity (*p* = 0.64).

After seven days of direct exposure, mitochondrial activity was still increased in the NiTi-1 and NiTi-2 samples in comparison to the untreated cells (Figure 6), with higher proliferation on NiTi-2, but, again, without a statistically significant difference between the groups NiTi-1 and NiTi-2 (*p* = 0.168). The ANOVA results and post hoc comparison tests showed that there was a statistically significant increase in mitochondrial activity between the NiTi-1 samples (mean = 114.44) and the control (mean = 100) (*p* < 0.01) and between the NiTi-2 samples (mean = 114.42) and the control (mean = 100) (*p* < 0.01).

SEM images of fibroblasts grown on the NiTi-1, NiTi-2, and control samples are given in Figure 7. Fibroblasts were created a thick cellular layer over the whole surface of the alloys.

## 4. Discussion

In the present study we compared the biofunctional properties of nickel–titanium alloys obtained by two different production processes. Contrary to conventional casting, continuous casting was previously reported to be a production process that was able to produce stands of relatively small diameter with an acceptable surface quality [17,25,26]. Even though the correlation between production processes and material properties has been proven in numerous studies [27,28], there is still no clearly defined consensus on the accepted model for the production of these alloys for application in dentistry.

From the perspective of the specimens’ chemical composition, it is evident that there is a difference between NiTi-1 and NiTi-2 in nickel and titanium content, which is the consequence of different production processes. Classical casting into ingots requires remelting in a vacuum furnace, due to Ti ions’ tendency to oxidize. Additionally, in this procedure, it is necessary to ensure appropriate mixing conditions for both components. High equilibrium conditions are usually achieved during remelting, with the formation of a practically homogeneous alloy, according to the NiTi phase diagram. In order to achieve the required dimensions for the usage in dentistry, nitinol products obtained by classical casting into ingots, have to undergo secondary fabrication methods (thermo-mechanical treatments). During rolling or drawing of the material, large deformations occur, as confirmed by microstructural analysis. On the other hand, a significantly lower mass (20 kg) of the nitinol alloy is used within the single procedure of continuous casting. The higher nickel and lower titanium content in NiTi-2 alloys is the consequence of mixing conditions of the melt. Namely, vacuum induction melting in the crucible is performed at a medium frequency (f = 8000 Hz) and additional mixing of Ni and Ti components would probably be necessary with use of lower frequencies (f = 2000 Hz). Even though continuous casting technology results in the different chemical contents of the Ni and Ti components, this fact does not negatively affect the biological characteristics of the obtained alloy. Additionally, since there is no need for secondary production processes, the microstructure of the continuous cast specimens is appropriate and better in comparison to the classical cast alloy.

Regarding the mechanical properties of nickel–titanium alloy, the hardness should be adapted to their clinical purpose. Given that nitinol is used predominantly in orthodontics for archwires, high hardness values are desirable. The problem with components of orthodontic arches with low hardness values is that they might compromise the transfer of torque force from an activated archwire to the bracket, which can cause possible plastic deformation of the wings [29]. Our study demonstrated a higher hardness of NiTi-2 in comparison to the NiTi-1 alloy. This can be explained by a higher nickel content, and/or iron content, as shown by EDX/XRF analysis, which is in agreement with previous reports [25,26]. An additional EBSD analysis revealed that there were two phases in the microstructure of NiTi-1 samples (NiTi-cubic and Ni_3_Ti hexagonal), while an additional NiTi_2_-cubic phase was detected in the NiTi-2 microstructure. It can be concluded that this phase was able to increase the hardness. It could also be concluded that the finer the microstructure, the higher the hardness.

Immersion tests performed for seven days estimated the quantity of Ni and Ti ions released in pH neutral and acidic environments. Different concentrations of released ions were only detected in the acidic environment. NiTi-1 and NiTi-2 alloys in an artificial saliva solution, which has a neutral pH, showed no significant difference in the concentration of released ions, nor were these amounts of clinical significance. These results were expected, considering that the good corrosion resistance in pH neutral solutions of nickel–titanium alloys is well documented [9,30,31]. Maximum allowable doses of nickel ions for humans are 0.5 μg/cm^2^/week [32]. This is 10 times higher than the results that we obtained in the artificial saliva. Our results indicate that the nickel ion release in artificial saliva was 0.04 μg/cm^2^/week from NiTi-2 samples and 0.05 μg/cm^2^/week from NiTi-1.

However, various studies have demonstrated lower corrosion resistance of these alloys in acidic solutions and chloride-containing environments [31,33,34]. In general, it is well documented that the corrosion potential increases as the pH value decreases [33,34,35]. The acidic environment had a significant effect on increasing the release of toxic ions from both groups of samples. However, the Ni ion release was two times lower, and the Ti ion release was three times lower from the NiTi-2 than from the NiTi-1 samples. Even though the oral cavity pH is predominantly neutral, changes towards the acidic environment may happen, but are short in duration, depending mainly on the consumption of specific foods. However, the duration of the immersion tests both in neutral and acidic environments should be extended in future experiments.

After the immersion test was performed in lactic acid, a surface oxide layer was formed in both groups of samples, NiTi-1 and NiTi-2. The results obtained with the FIB cross-section showed similar thicknesses of the oxide layer formed on samples’ surfaces, even though ICP analysis showed different concentrations of metal ions released from the samples. The lower values of the standard deviations for the depth of the oxide layer in the NiTi-2 samples also indicated a greater data consistency in continuous casting, even though the thickness of the corrosive layer was almost the same in both samples.

In terms of Ni and Ti ion release, the results from both groups of samples showed that the release of nickel ions was higher than of titanium ions, and this supports the previous studies [36,37]; it can be explained by the fact that, after the particular dissolution of the surface TiO_2_layer, the inner layer, which is an Ni-rich layer, is exposed [37,38]. According to the literature data, the surface characteristics of materials act like ion diffusion barriers [27,28,38]. It is hard to obtain such a surface integrity without secondary surface modification processes and, to our knowledge, this is one of the first studies that compares classical and continuous casting as the primary manufacturing process. The present research established that the NiTi-2 alloy, which has a more stable microstructure with smaller grains and a higher grain number, is more biologically resistant. It also showed that Ni ion release was not proportional to the content of nickel in the alloy samples, a finding that is in accordance with previous investigations [34,39].

The present study clearly demonstrated the biocompatibility of nickel–titanium alloy manufactured by continuous casting. To be able to perform biocompatibility testing on appropriate cells, it is necessary to know the different cellular behaviors for the different material surface properties. Fibroblasts are known to be present in almost all types of tissues and to play an important role in tissue inflammatory reactions. Moreover, these cells prefer smoother surfaces, showing better adhesion to them [27]. Accordingly, we prepared samples by the machine polishing of their surfaces. Generally, the mitochondrial activity of cells grown directly on the tested samples was greater at both points in time compared to the control cells with a proliferation tendency during the observation time. However, further investigations are needed for the evaluation of the potential genotoxic effects of these alloys.

## 5. Conclusions

On the basis of the present study, it can be concluded that there is a significant difference in the biofunctional properties between nickel–titanium alloys obtained by classical and by continuous casting. Nickel–titanium alloys obtained by continuous casting have a more stable microstructure, higher hardness, and better resistance. They also have an additional NiTi_2_-cubic phase in their structures that contributes to greater stability and biocompatibility.

## Figures and Tables

**Figure 1 molecules-27-01960-f001:**
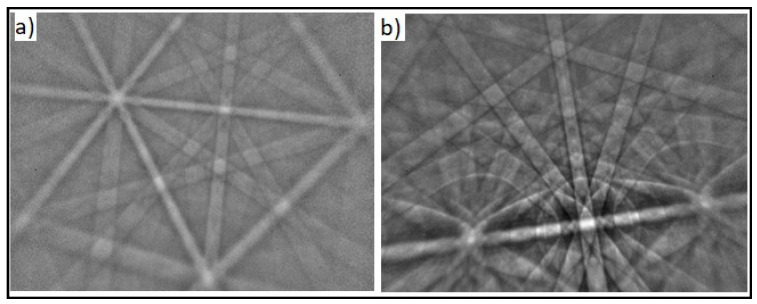
SEM image with the EBSD analysis of an NiTi-1 sample (0.5 um scale): (**a**) NiTi cubic phase; (**b**) Ni_3_Ti hexagonal phase.

**Figure 2 molecules-27-01960-f002:**
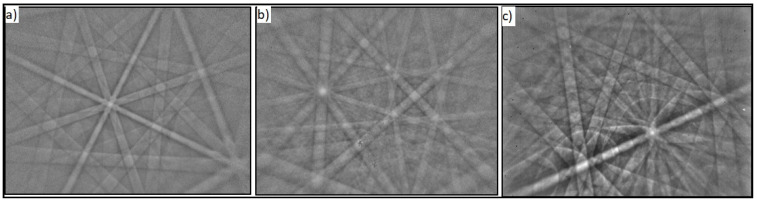
SEM image with the EBSD analysis of an NiTi-2 sample (0.5 um scale): (**a**) NiTi cubic phase; (**b**) NiTi_2_ cubic phase; (**c**) Ni_3_Ti hexagonal phase.

**Figure 3 molecules-27-01960-f003:**
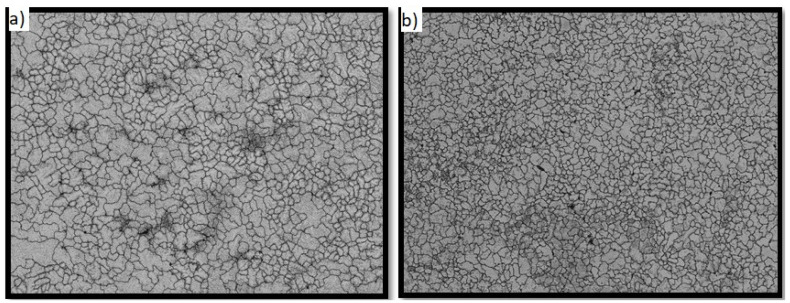
Micrographs of grains and boundaries (500 um scale): (**a**) NiTi-1 sample; (**b**) NiTi-2 sample.

**Figure 4 molecules-27-01960-f004:**
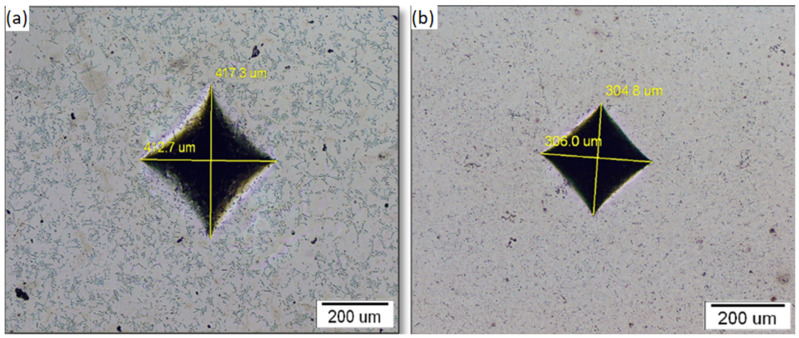
Indentation imprint: (**a**) NiTi-1 sample; (**b**) NiTi-2 sample.

**Figure 5 molecules-27-01960-f005:**
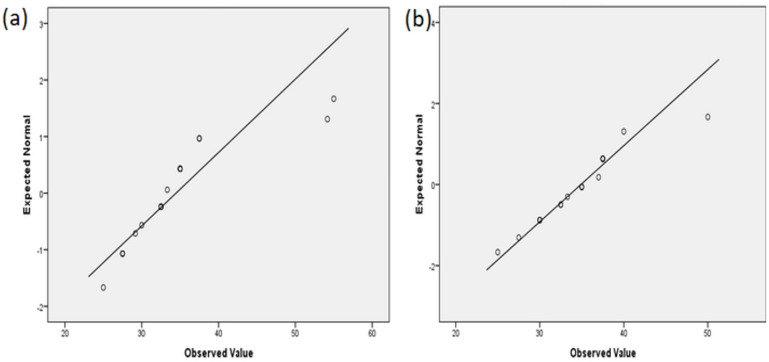
Normal Q−Q plot of the oxide layer thickness: (**a**) NiTi-1 sample; (**b**) NiTi-2 sample.

**Figure 6 molecules-27-01960-f006:**
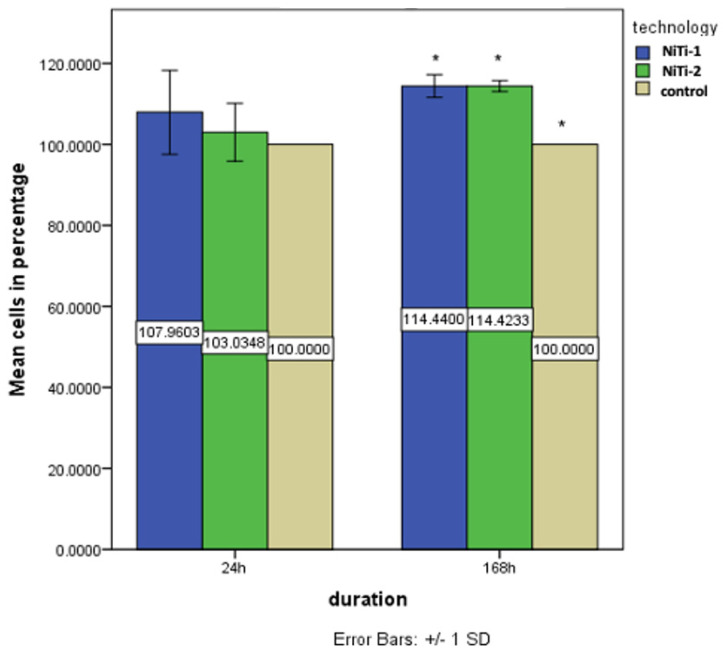
Cell viability for the three groups (NiTi-1, NiTi-2 and control) at 24 h and 168 h. (* *p* < 0.05).

**Figure 7 molecules-27-01960-f007:**
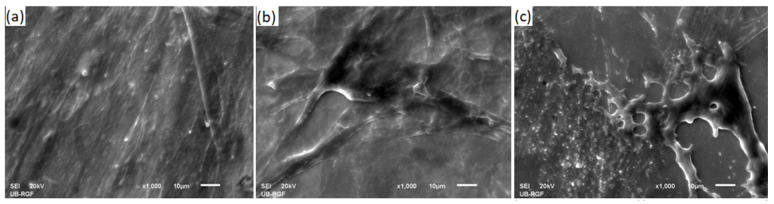
SEM image of fibroblast cells cultured on samples: (**a**) NiTi-1; (**b**) NiTi-2; (**c**) control sample (a sterile glass disc).

**Table 1 molecules-27-01960-t001:** Composition of the NiTi-1 and NiTi-2 samples using XRF and EDX analyses.

Element (wt.%)	NiTi-1	NiTi-2
XRF:		
Nickel	55.20	62.50–63.60
Titanium	44.80	35.90
Iron	Not detected	1.40
EDX:		
Nickel	47.62	63.2
Titanium	52.38	34.4
Iron	Not detected	1.53

**Table 2 molecules-27-01960-t002:** ASTM analysis results.

Sample	Grain Number G	Number of Grains per mm^2^	Mean Number of Intersections (mm)
NiTi-1	5	256	0.0527
NiTi-2	7	1024	0.0234

**Table 3 molecules-27-01960-t003:** Descriptive statistics for hardness measurements for the NiTi-1 and NiTi-2 samples.

	NiTi-1	NiTi-2
Mean	317	624
St. Dev.	23	20
Min	289	586
Max	357	644
N	8	8

Note: N = Number of measurements.

**Table 4 molecules-27-01960-t004:** ICP analysis results in µg/cm^2^ after 7 days of the immersion of NiTi-1 and NiTi-2 samples in artificial saliva (pH 6.5) and lactic acid (pH 2.3).

	Sample	Al	Cu	Ni	Ti
Blank solution		0.06	0.08	<0.01	<0.01
Artificial	Niti-1	Not detected	Not detected	0.05	<0.01
Saliva pH 6.5	NiTi-2	Not detected	Not detected	0.04	<0.01
Blank solution		0.05	0.37	0.02	0.01
Lactic	Niti-1	0.01	0.37	2.33	2.17
acid pH 2.3	NiTi-2	0.06	0.45	1.2	0.63

**Table 5 molecules-27-01960-t005:** The descriptive statistics of the oxide layer thickness (in nm) for NiTi-1 and NiTi-2 samples.

	NiTi-1	NiTi-2
Mean	34	35
St. Dev.	8	5
Min	27	25
Max	55	50
N	20	20

Note: N = number of measurements.

## Data Availability

The data presented in this study are available on request from the corresponding author. The data are not publicly available as they are part of a yet undefended PhD thesis.

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
