# Peer review of "Experimental Investigation of the Biofunctional Properties of Nickel–Titanium Alloys Depending on the Type of Production"

_molecules, 2022, doi:10.3390/molecules27061960_

Round 1

Reviewer 1 Report

The article is of interest for practical dentistry. However, it needs major editing. The following are questions that require clarification.

  1. Page 6. Figure 1.

“Figure 1 presents the optical microstructure of imprint where the length of the diagonal of the impression was created as a negative (copy) of the indenter. As can be seen, the impression of NiTi-1 is smaller than in the NiTi-2 sample.” Why? Figure 1: 4173 um is smaller than 3048 um? Please, explain.

  1. Page 6. 3.3. ICP analysis of solutions after immersion testing

The authors performed ICP analysis of solutions after immersion testing. They showed the presence of released Ni, Ti, Cu, Al ions, measured by the ICP-MS analysis, for both NiTi-1 and NiTi-2 samples in solutions of artificial saliva and lactic acid (Table 4.). However, they did not compare their results with the maximum allowable concentration of these ions for humans. Without such a comparison, the results obtained become meaningless.

  1. Page 7.

The immersion of NiTi-1 and NiTi-2 samples in solutions of artificial saliva and lactic acid did not exceed 7 days. However, products made from the investigated alloys usually work for many months. How will the investigated alloys behave during long-term use? What are the predictions? The article says nothing about this. Such research needs to be done.

This remark also applies to Section 3.5. biocompatibility results

  1. Page 7. 3.4. FIB cross-section analysis

Figures 2 and 3 are not informative. Moreover, the data of these figures are duplicated in Figure 4 and in Table 5. I recommend removing Figures 2 and 3 from the manuscript.

  1. Page 11. Lines 362-368.

Authors approves, that a smaller concentration of metal ions released from the NiTi-2 samples in the lactic acid solution is due to by better integrity and homogeneity of the oxide layer. Such a conclusion must be confirmed, for example, by microscopic studies. However, there are no such data in the manuscript.

  1. The article presents experimental results without explaining the mechanism of ongoing processes and without scientific justification. A deeper and more thorough analysis of the experimental data is needed.
  2. There are many repeats in the text of the article. The manuscript requires careful editing. Only after that can be made a decide about publishing the article in Molecules.

    The article is of interest for practical dentistry. However, it needs major editing. The following are questions that require clarification.

                            1.  

Reviewer 2 Report

Authors have present a research work with sufficient scientific interest to be publish, however there are many incomplete experimental and results analysis that force me to reject this paper at its present form. Moreover, author seems to limits their contribution to corroborate previous results.

The following aspect must be taken under deep consideration by the authors:

  1. Abstract: Please clearly indicate the manufacturing technique by the term commercial if you are focused on this matter.
  2. Line 62: Please include a list of dental pieces made with NiTi
  3. Line 84: Describe in greater detail the term conventional fabrication methods
  4. Line 108: The casting method must be mentioned previously in the text. Moreover, the main difference between continuous and conventional casting must be highlighted. Technically, both undergo a solidification process.
  5. Line 127: Etching time seem excessive, could you mention a standard procedure for this? ASTM, ASM, ISO, etc
  6. Line 189-193: This paragraph is almost the same as previous one. Check and delete accordingly.
  7. Line 229: There is no correlation of the hardness value with the microstructure. The grain size must be calculated and the phase analysis from figure 1 must be included.
  8. Line 230: Descriptive
  9. Line 237: presents the
  10. Line 237: Please use the term indentation instead of imprint.
  11. Line 270: I cannot recognize the thickness measured in Figure 2. If the thickness varies from 25 to 55 nm I cannot see the difference on the SEM images.
  12. Line 336-340: I respectfully disagree, without the grain size of the NiTi phase and the among of Ni3Ti phase. No strong dependency over the manufacturing process can be established. I recommend performing an XRD test to determine the Ni3Ti phase and to run a metallographic characterization to determine the NiTi grain size.
  13. Line 362-363: Once again, if authors do not determine the grain size and the phase distribution, cannot establish a reason for the different or similar corrosion behaviour.

Reviewer 3 Report

In this manuscript the chosen properties of NiTi alloy produced by different technologies were presented.

  • The introduction is properly prepared
  • The experimental section I also mostly adequately described. However, some improvement should be made.

NiTi-1 – manufacturer and other data are is known? We should know more about it.

“NiTi-2, were produced by continuous casting in the form of rods, according to the briefly described methodology” – please give more information’s.

  • You write in materials and methods section that microstructure studies have been carried out. Why are the results of these studies were not presented and described?
  • From the analysis of the chemical composition, it appears that the two materials purchased/obtained differ significantly in their chemical composition. We can say that you have two alloys. The question is, where does this difference come from, and whether the variation in all presented results is due to the difference in chemical composition or deference the manufacturing process used (as you declare in title)? Wouldn't it be better to try to obtain test samples whose basic chemical composition will be similar to each other? In my opinion it is  problematic  in the context of the title…
  • Hardness values should be rounded to the nearest unit (no decimal places). I don't see any point in showing an indentation (fig. 1).
  • Oxide layer – table 5 should be rounded to the nearest unit (no decimal places).

Reviewer 4 Report

Title

Experimental investigation of the biofunctional properties of Nickel-titanium alloy depending on the type of production

The aim of this study is to investigate the influence of alloy fabrication (continuous or conventional casting)on the biofunctional properties of Nickel-titanium alloy in terms of hardness, ion release from the surface in a testing medium, oxide layer formation and biocompatibility.

The manuscript is well written. I have some minor comments/suggestions:

Abstract

Nickel-titanium alloys used in Dentistry have a variety of mechanical, chemical, and biofunctional properties that are dependent on the manufacturing process. The aim of this study was to compare the mechanical and biofunctional performances of a Nickel-titanium alloy produced by the continuous casting method (NiTi-2) with commercial Nitinol (NiTi-1). The chemical compositions of the tested samples were analyzed using an energy dispersive X-Ray analysis (EDX) and X-ray fluorescence (XRF). As part of the mechanical properties, the hardness on the surface of samples was measured with the static Vickers method. The release of metal ions (Ni, Ti) in artificial saliva (pH 6.5) and lactic acid (pH 2.3) was measured using a static immersion test. Finally, the resulting corrosion layer was revealed by means of a Scanning Electron Microscope, which allows the detection and direct measurement of the formatted oxide layer thickness. To assess the biocompatibility of the tested Nickel-titanium alloys samples an MTT test was performed, evaluating the fibroblast cellular proliferation on direct contact with samples after 24 h and 168 h. The obtained data were analyzed with the IBM SPSS Statistics v22 software. EDX and XRF analyses showed in the NiTi-2 sample a higher presence of Ni. Also, in the NiTi-2 higher hardness was measured. Immersion test in artificial saliva with pH 6.5 after 7 days, did not induce significant ion release in neither groups of samples (NiTi-1 and NiTi-2). The acidic environment had a significant effect on increasing the release of toxic ions. Ni ions release was two times lower, and Ti ions release was three times lower from the NiTi-2. A comparison of the cells` mitochondrial activity between the NiTi-1 and NiTi-2 groups did not show a statistically significant difference. This study revealed that the biofunctional properties of Nickel-titanium alloy depend on its chemical composition and the type of manufacturing process.

Abstract: Suggest to add 1 sentence on the discussion of major findings

  1. Material and Methods

2.1. Samples preparation

2.2. Determination of chemical composition

2.3. Hardness

2.4. Immersion testing and ICP-MS analysis

2.5. FIB cross-section of the immersed samples

2.6. In vitro determination of biocompatibility

2.7. Statistical analysis

  1. Results

3.1. Chemical composition (EDX and XRF analyses)

3.2. Hardness

3.3. ICP analysis of solutions after immersion testing

3.4. FIB cross-section analysis

Comment: Method and Results sections are well described. 2.2. Determination of chemical composition: Suggestion to revise subheading 2.2 because this section also includes SEM

Table 1. Chemical composition of the artificial saliva with a 6.5 pH value.

Table 2. Composition of the NiTi-1 and NiTi-2 samples using XRF and EDX analyses

Table 3. Desciptive statistic for hardness measurments for the NiTi-1 and NiTi-2 samples

Table 4. ICP analysis results in μg/cm2for 7 days: for NiTi-1 and NiTi-2 samples in artificial saliva 248 (pH 6.5) and lactic acid (pH 2.3)

Table 5. Descriptive statistics of the oxide layer thickness (in nm) for NiTi-1 and NiTi-2 samples.

Suggestion: Omit Table 1

Figure 1. Indentation imprint: (a) NiTi-1 sample; (b) NiTi-2 sample

Figure 2. FIB cross-section of NiTi-1 after the immersion test in lactic acid at four different meas-273 uring segments: (a) In the first measuring segment the thickness varied between 25-54.17 nm; (b) In 274 the second measuring segment the thickness of the oxide layer was in the range of 32.50-37.50nm;

Figure 3. FIB cross-section of NiTi-2 after the immersion test in lactic acid at four diffierent meas-282 uring segments: (a) In the first measuring segment the thickness varied between 32.50-37.50 nm; (b) 283 In the second measuring segment the thickness of the oxide layer ws in the range of 25-32.50nm; (c) 284 In the third measuring segmentthe oxide layer thickness was between 37.50 and 50nm; (d) In the 285 fourth measuring segment the oxide layer thickness was between 30 and 40nm.

Figure 4. Normal Q-Q Plot of the oxidelayer thickness: (a) NiTi-1 sample; (b) NiTi-2 sample

Figure 5. Graph of cell viability for three groups (NiTi-1, NiTi-2 and contol) at 24 and 168 hours.

Figure 6. SEM image of fibroblast cells cultured on samples: (a) NiTi-1 sample; (b) NiTi-2 sample; 319 (c) Control sample (a sterile glass disc).

Figure: try to streamline and combine some of these figures.

typo

and the continuous casting method seems to pro-403 duce NiTi rods with biological properties suitablefor use in Dentistry.

Table 3. Desciptive statistic for hardness measurments for the NiTi-1 and NiTi-2 samples

25-32.50nm; (c) 284 In the third measuring segmentthe oxide layer thickness was between 37.50

Round 2

Reviewer 1 Report

The authors carefully considered the comments made, edited the text, added the necessary explanations. All this improved the manuscript.
However, minor editing of Figures 1-3 is required. You must specify the scale or magnification.

The corrected version of the article may be published in "Molecules".

Reviewer 2 Report

The authors have properly addressed all my comments and suggestions Congratulations, this paper is now in much better shape for publication.

Please run a spell check to improve the quality of this manuscript

Reviewer 3 Report

Dear Authors, thank you for  improvements. Please note that question 4 was intended to provoke you to make changes to the manuscript. 
Such a matter should not be left without any comment. Thank you very much for the explanations, but the point is that this doubts should not exist - the only way for this to happen is to describe it in article, what you wrote in ansvers. Remember that your work is supposed to have both scientific and educational value, and this matter without explanation and lacks in methodology section was simply strange. Therefore, I believe that  considerations on this subject should be included in the discussion - so please improve it.
